# Macrophage-derived IGF-1 protects the neonatal intestine against necrotizing enterocolitis by promoting microvascular development

Xiaocai Yan [1,2], Elizabeth Managlia[1,2], You-Yang Zhao[3,4], Xiao-Di Tan[2,5] & Isabelle G. De Plaen [1,2✉]

Necrotizing enterocolitis (NEC) is a deadly bowel necrotic disease of premature infants. Low levels of plasma IGF-1 predispose premature infants to NEC. While increasing evidence suggests that defective perinatal intestinal microvascular development plays a role in NEC, the involved mechanism remains incompletely understood. We report here that serum and intestinal IGF-1 are developmentally regulated during the perinatal period in mice and decrease during experimental NEC. Neonatal intestinal macrophages produce IGF-1 and promote endothelial cell sprouting in vitro via IGF-1 signaling. In vivo, in the neonatal intestine, macrophage-derived IGF-1 promotes VEGF expression and endothelial cell proliferation and protects against experimental NEC. Exogenous IGF-1 preserves intestinal microvascular density and protects against experimental NEC. In human NEC tissues, villous endothelial cell proliferation and IGF-1- producing macrophages are decreased compared to controls. Together, our results suggest that defective IGF-1-production by neonatal macrophages impairs neonatal intestinal microvascular development and predisposes the intestine to necrotizing enterocolitis.

[1] Division of Neonatology, Department of Pediatrics, Ann & Robert H. Lurie Children's Hospital of Chicago, Northwestern University Feinberg School of Medicine, Chicago, IL, USA. [2] Center for Intestinal and Liver Inflammation Research, Stanley Manne Children's. Research Institute, Ann & Robert H. Lurie Children's Hospital of Chicago, Northwestern University Feinberg School of Medicine, Chicago, IL, USA. [3] Program for Lung and Vascular Biology, Stanley Manne Children's. Research Institute, Ann & Robert H. Lurie Children's Hospital of Chicago, Chicago, IL, USA. [4] Departments of Pediatrics, Pharmacology and Medicine, Northwestern University Feinberg School of Medicine, Chicago, IL, USA. [5] Division of Gastroenterology, Department of Pediatrics, Ann & Robert H. Lurie Children's Hospital of Chicago, Northwestern University Feinberg School of Medicine, Chicago, IL, USA. ✉email: isabelledp@northwestern.edu

Necrotizing enterocolitis (NEC) is a leading cause of morbidity and mortality amongst premature infants and is characterized by intestinal mucosal inflammation and necrosis, leading to overwhelming sepsis[1]. NEC is considered a multifactorial disease where an immature immune response, an immature mucosal barrier, and abnormal intestinal microbiota play a role[2–4]. During the normal neonatal period, the intestine undergoes a great expansion of its microvascular network[5] and our recent evidence suggests that defective intestinal microvascular development significantly contributes to NEC[6]. The expression of the proangiogenic factor vascular endothelial cell growth factor-A (VEGF) is decreased in human intestinal tissues obtained from infants with NEC[7]. VEGF polymorphisms have been associated with NEC[8] and VEGF gene DNA methylation of intestinal cells has been shown to precede NEC[9]. In neonatal mice, inhibition of VEGF receptor 2 (VEGFR2), the main receptor for VEGF, decreases intestinal vascular endothelial cell proliferation and microvasculature density and increases the incidence of experimental NEC[10]. Restoring VEGF production preserves intestinal endothelial cell proliferation and decreases the incidence of severe NEC in neonatal mice[11]. However, factors that promote microvasculature development in the intestine during the perinatal period remain incompletely understood.

Insulin-like growth factor-1 (IGF-1) is a growth hormone that plays an important role in intrauterine and postnatal growth[12]. It is produced by the placenta[13] and the liver and is present at high concentration in the colostrum, slowly declining in human breast milk over the first 6 months of life[14–16]. It is bound to IGF-binding proteins in the plasma that regulate its stability and its biological availability to peripheral tissues[17]. In addition, IGF-1 is locally produced in several tissues, having paracrine and autocrine effects. Low levels of plasma IGF-1 in premature infants are associated with an increased risk of NEC, retinopathy of prematurity (ROP), and broncho-pulmonary dysplasia (BPD)[18]. Recently, exogenous rhIGF-1/BP3 has been shown to prevent pulmonary hypertension in experimental BPD[19] and to decrease the incidence of severe NEC in a pig model[20]. Although IGF-1 has been shown to promote proliferation and to decrease apoptosis of enterocytes[21], its role on the perinatal intestinal microvasculature remains unclear.

Neonatal intestinal macrophages (Mϕ) are mostly originated from yolk sac macrophages and fetal liver monocytes and seeded before birth[22,23]. Embryonic macrophages are necessary for the development of the vasculature in embryonic tissues[24–26] by acting as cellular chaperones to promote vascular anastomosis[27]. In the embryonic heart, yolk sac-derived macrophages have been shown to promote coronary development via IGF-1[24] and IGF-1 induces VEGF secretion in embryonic stem cells[25]. However, whether and how neonatal macrophages promote the development of the intestinal mucosal microvasculature remains unknown.

In this study, we first defined the developmental expression of IGF-1 in the mouse perinatal intestine and determined that intestinal IGF-1 is significantly decreased in the intestine prior to NEC development in a mouse model. We found a critical role for macrophages in promoting intestinal microvascular development, which are juxtaposed to endothelial cells in the neonatal intestinal villi in vivo. Secondly, using both in vitro and in vivo approaches, we identified IGF-1 as a novel mechanism by which macrophages promote intestinal microvasculature development in neonatal mice to prevent NEC. Third, we confirmed the relevance of our findings in human NEC.

## Results

### IGF-1 expression is developmentally regulated during the perinatal period in mice

Serum IGF-1 levels from mouse fetuses and pups of different ages were measured by enzyme-linked immunosorbent assay (ELISA) and intestinal IGF-1 and IGF-1 receptor (IGF-1R) expression were assessed by western blot analysis. We found that serum IGF-1 levels decreased at birth with a nadir at around day 1 of life, then gradually increased during the first 3 weeks of life (Fig. 1a). Similarly, intestinal expression of IGF-1 and of its receptor IGF-1R sharply decreased at birth (Fig. 1b). However, although during the first week of life intestinal IGF-1 and IGF-1R expression increased transiently, peaking at ~7 days for IGF-1 and at 7–14 days for IGF-1R, respectively, it decreased to low levels by day 21 of life (Fig. 1b).

### IGF-1/IGF-1R expression is decreased prior to experimental NEC

To determine whether experimental NEC affects IGF-1/IGF-1R expression, serum IGF-1 levels and intestinal IGF-1 and IGF-1R were analyzed after initiation of the NEC protocol. Although no significant changes were noted at 6, 8, and 10 h (results not shown), we found serum IGF-1 to be decreased at 12 h of the NEC protocol, with a further decrease at 24 h (Fig. 2a). Intestinal IGF-1 and IGF-1R were both decreased as early as 8 h after experimental NEC protocol initiation (Fig. 2b, c) and remained low at 72 h into NEC (not shown), which is earlier than IGF-1 changes observed in the serum. To assess the source of IGF-1 production in the neonatal intestine, intestinal epithelial cell, macrophage, and endothelial cell fractions of 24 h-old dam-fed pups were assessed for pro-IGF-1 (17 kD)[28] by western blot analysis. Intestinal macrophages and endothelial cells produced significantly higher amounts of IGF-1 compared with intestinal epithelial cells (Fig. 2d).

### Neonatal intestinal macrophages express IGF-1 and promote endothelial cell sprouting in vitro via an IGF-1-dependent mechanism

We found macrophages to be juxtaposed to endothelial cells in the intestinal villi of neonatal mice (Fig. 3c and Supplementary Fig. 1). Interestingly, their morphology was different during NEC compared to dam-fed controls (Supplementary Fig. 1). In the embryo, yolk sac-derived macrophages promote coronary development via IGF-1[24]. Whether neonatal intestinal macrophages promote intestinal microvasculature development during the neonatal period via a similar mechanism remains unknown. To study whether neonatal intestinal macrophages promote endothelial cell sprouting in vitro, red-fluorescent intestinal endothelial cells [isolated from neonatal mT/mG mouse lamina propria (LP)] and CD11b+ myeloid cells (from neonatal Cx3cr1-GFP reporter mouse LP) were cultured separately or together in Matrigel with growth factor-containing culture media (LL-0005, Lifeline Cell Technology) and endothelial cell sprouting was assessed (Fig. 3a, b). In contrast to human umbilical vein endothelial cells[29], no sprouting could be observed in neonatal intestinal endothelial cells cultured separately (Fig. 3a top row). However, endothelial cell sprouting could be readily observed within 2 days, with maximal growth reached at around 72 h when co-cultured with neonatal intestinal CD11b+ myeloid cells (Fig. 3a bottom row, Fig. 3b). In addition, when cultured together with CX3CR1+ macrophages in a 24-well plate in growth factor-deprived media (LL-0004, Lifeline Cell Technology) for 48 h, endothelial cell proliferation was significantly higher than when endothelial cells were cultured separately (Fig. 3d, e—Gating strategy in Supplementary Figure 2) and the number of endothelial cells was significantly increased (Fig. 3f).

Next, we assessed whether neonatal intestinal macrophages and endothelial cells affect their respective expression of IGF-1 and IGF-1R when cultured together for 48 h. Interestingly, we found increased intracellular IGF-1 expression in CX3CR1+ cells co-cultured with endothelial cells, compared to CX3CR1+ cells

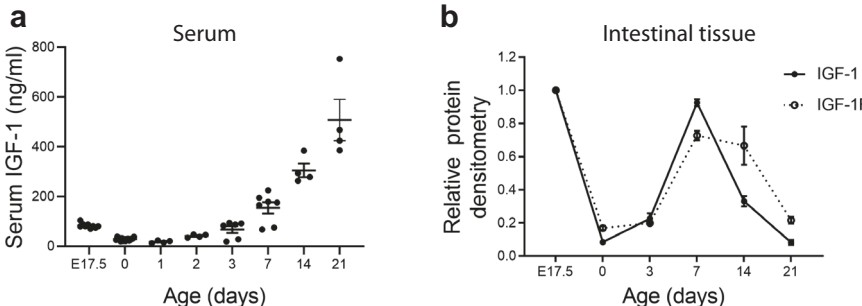

**Fig. 1 Serum and intestinal IGF-1 are developmentally regulated during the perinatal period in mice.** Pups from each litter were randomly assigned to different time-points. Serum and intestinal tissues were collected from mice at embryonic day 17.5 (E17.5), day of life 0, 1, 2, 3, 7, 14, and 21. **a** IGF-1 serum level was measured by ELISA, $n = 4$–12/time point (exact $n$ at each time point see dots in the panels and Source Data file.), data represent results of three experiments combined (mean ± SEM). **b** Intestinal IGF-1 and IGF-1R were examined by western blot analysis. Data represent the results of three experiments. $n = 2$/time point for every three separate experiments (mean ± SEM). One of the three experiments did not include E17.5 time point. Gel images are included in Source Data file.

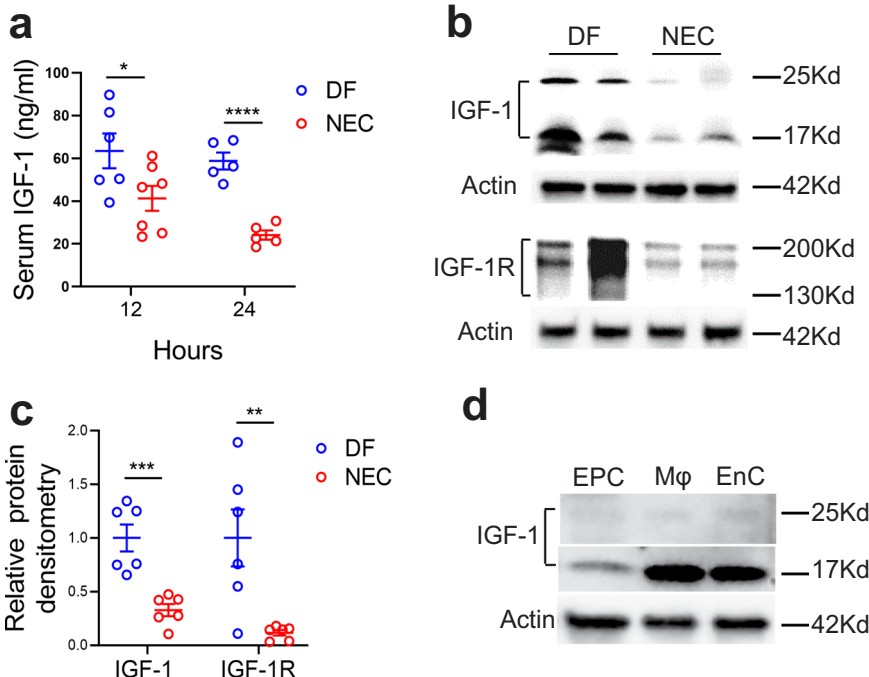

**Fig. 2 IGF-1 and IGF-1R expression are decreased prior to experimental NEC.** Day 1 neonatal mice were left with the dam (DF) or submitted to the experimental NEC model. **a** 12 and 24 h later, their serum was obtained, and IGF-1 level was determined by ELISA. $n = 5$–7/group (see dots in panels and Source Data file for exact $n$ number). **b**, **c** Intestinal tissues were collected from NEC pups 8 h after protocol initiation or from DF controls, and tissue lysates were submitted for western blot analysis to examine IGF-1 and IGF-1R expression. **b** Typical blots are shown. **c** Protein densitometry analysis, $n = 6$/group. Data represent results of three experiments combined (**a**, **c**, mean ± SEM) and $P$ values were calculated using multiple $t$ tests (**a**, **c**). *$p < 0.05$, **$p < 0.01$, ***$p < 0.001$, ****$p < 0.0001$. **d** IGF-1 expression from the intestinal epithelial cells (EPC), macrophages (Mφ), and endothelial cells (EnC) of 24 h-old dam-fed pups were assessed by western blot. Data represent two separate experiments and each experiment includes 9 and 12 intestines pooled, respectively. Source data are provided as a Source Data file.

cultured separately, suggesting that macrophages sense the presence of endothelial cells to produce higher levels of IGF-1 (Fig. 3g, h). However, co-culture status only marginally increased endothelial cell expression of IGF-1, which was not statistically significant (Fig. 3g, h). Also, the expression of the IGF-1 receptor by both cell types was not affected by co-culture status (not shown).

To define the role of IGF-1 on endothelial cell sprouting in vitro, endothelial cells and macrophages were co-cultured on Matrigel in the presence of either exogenous recombinant IGF-1 (100 ng/ml) and/or the IGF-1R inhibitor picropodophyllin (PPP) (500 nM), or of IGF-1-free control media alone (LL-0005 without

IGF-1). Endothelial cell sprouting was strongly enhanced by exogenous IGF-1 and was inhibited by PPP (Fig. 4a, b). Furthermore, exogenous IGF-1 failed to rescue endothelial cell sprouting when IGF-1 signaling was blocked by PPP (Fig. 4a, b).

**Macrophage-derived IGF-1 promotes intestinal vascular endothelial cell proliferation in vivo in neonatal mice and protects against experimental NEC.** To examine the role of macrophage-derived IGF-1 on intestinal vascular endothelial cell proliferation in vivo, transgenic mice that express Cre recombinase driven by the *Cx3cr1* promoter were bred with mice bearing

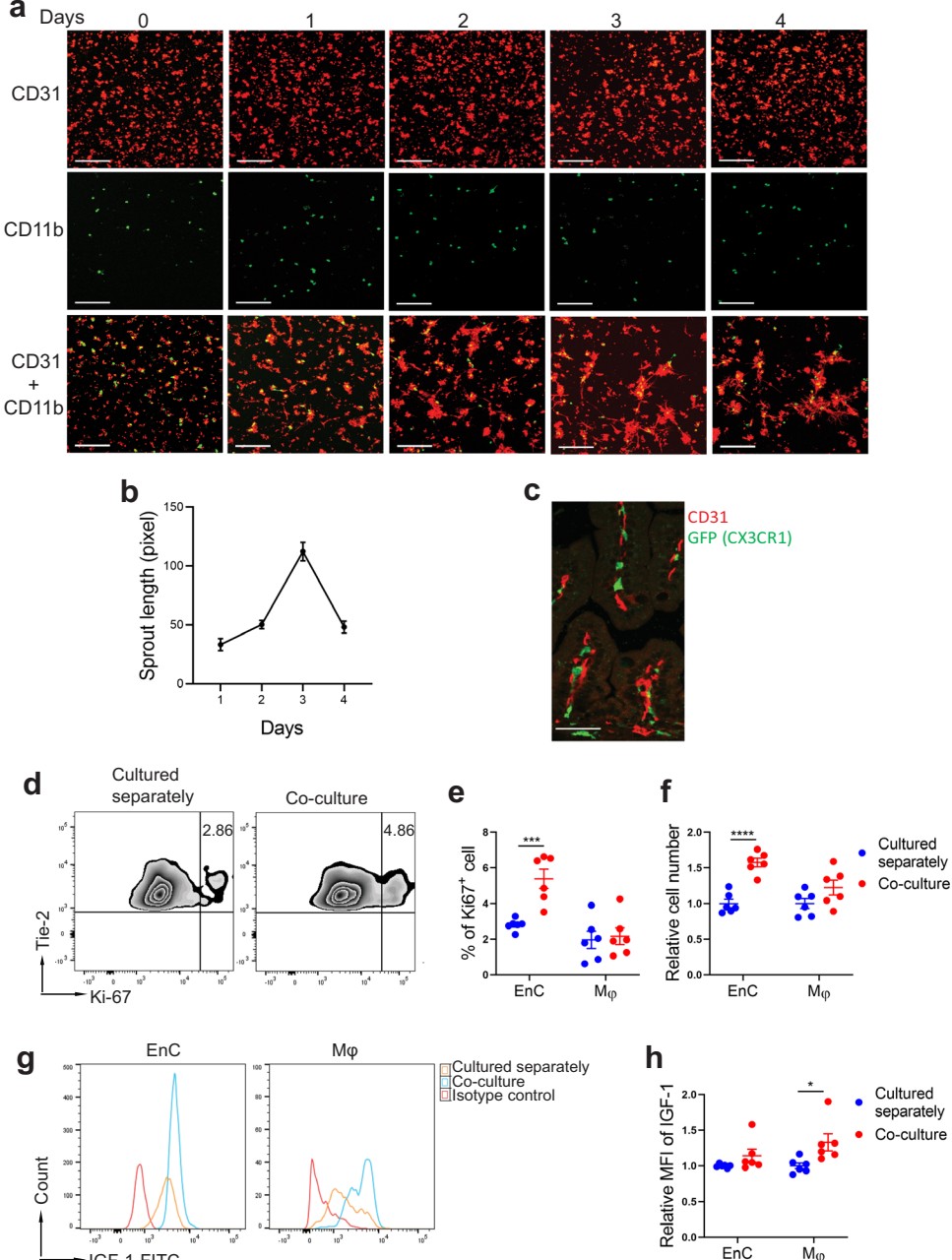

**Fig. 3 Neonatal intestinal macrophages promote endothelial cell sprouting/proliferation in vitro via IGF-1. a** $4 \times 10^4$ intestinal endothelial cells (from mT/mG mice—red) and $2 \times 10^4$ CD11b$^+$ intestinal myeloid cells (from CX3CR1-GFP mice—green) were cultured in duplicated wells separately or together in Matrigel and images were taken at day 1–4 of culture. scale bar: 100 μm. **b** When CD31 and CD11b were cultured together, endothelial cell sprout length per area was assessed over time (mean ± SEM). For each time point, 7–10 sprouts per area were measured. Sprouts were unmeasurable in individually cultured cells. **c** Small intestinal tissue of a 24 h-old CX3CR1-GFP reporter neonatal dam-fed pup was stained for CD31 (red) and GFP (CX3CR1 -green). Scale bar = 100 μm. **d–h** $5 \times 10^4$ intestinal CX3CR1$^+$ macrophages and $10 \times 10^4$ endothelial cells cultured separately or together in 24-well plates were collected at 48 h of culture and stained with anti-Tie-2, CX3CR1, Ki-67, IGF-1 antibodies or isotype controls for flow cytometric analysis. **d** Contour plots showing the percentage of proliferating (Ki-67$^+$) endothelial cells. **e** Percentage of proliferating (Ki-67$^+$) endothelial cells (Tie-2$^+$ CX3CR1$^-$ and macrophages (Tie-2$^-$CX3CR1$^+$). **f** Relative number of macrophages and endothelial cells in co-culture to their respective number when grown separately. **g** Histogram showing IGF-1 expression of neonatal intestinal endothelial cells and macrophages when cultured separately or together compared to isotype antibody control. **h** Mean fluorescence intensity of IGF-1 of the two cell populations when cultured separately or together. Data indicate two separate experiments combined (mean ± SEM for **e**, **f**, **h**), $n = 6$/group (**e**, **f**, **h**). See supplemental Fig. 3 for gating strategy for **d** and **g**. P values were calculated using multiple $t$ tests (**e**, **f**, **h**). *$p < 0.05$, ***$p < 0.001$, ****$p < 0.0001$. Source data are provided as a Source Data file.

loxP sites flanking exon 4 of the *Igf1* gene, to generate mice with IGF-1-deficient macrophages (Fig. 5a). We had confirmed that, in the neonatal intestine, CX3CR1$^+$ cells express CD11b, F4/80, CD64, but are negative for Ly6C, CD11c, and NK1.1 and are

therefore macrophages (Supplementary Figure 3). Litters with ~50% of the pups having IGF-1-deficient macrophages (*Igf1*$^{f/}$$^f$*Cx3cr1-Cre*$^{+/-}$, named *Igf1*$^{\Delta M\phi}$), and 50% having IGF-1-sufficient macrophages (*Igf1*$^{f/f}$ = controls) were generated.

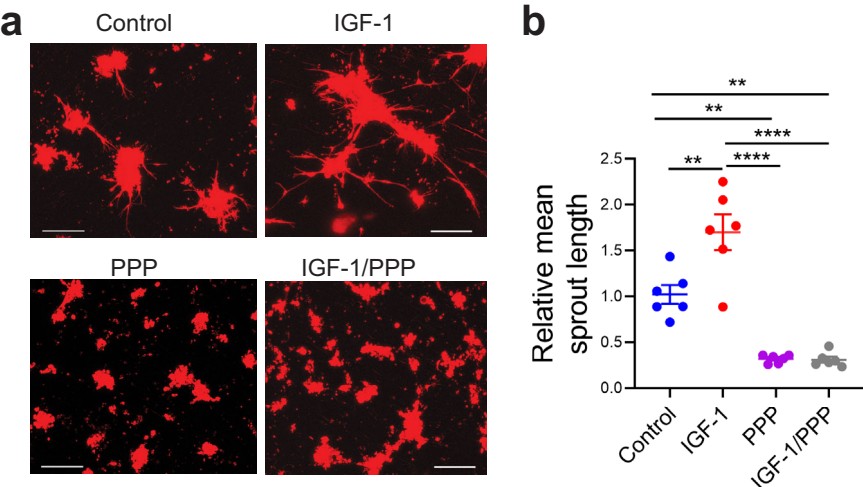

**Fig. 4 Neonatal intestinal macrophages promote endothelial cell sprouting in vitro via IGF-1. a, b** $4 \times 10^4$ mT/mG neonatal intestinal endothelial cells were co-cultured with $2 \times 10^4$ CX3CR1$^+$ WT neonatal intestinal macrophages for 4 days on Matrigel in IGF-1-free media (control) or in media containing IGF-1, the IGF-1 inhibitor PPP or both. **a** Representative images. **b** Graph represents mean sprout length per well relative to control. Data are the results of three separate experiments combined. $n = 6$ wells/group, each dot indicates the average length of 5–9 sprouts from a single culture well. $P$ values were calculated using one-way ANOVA followed by Turkey–Kramer multiple-comparison test. **$p < 0.01$, ***$p < 0.0001$. Source data are provided as a Source Data file.

Enriched CX3CR1$^+$ cells of $Igf1^{\Delta M\phi}$ mice isolated by magnetic column selection showed significantly lower IGF-1 expression compared to those of $Igf1^{f/f}$ controls when assessed by western blot (Fig. 5b, c). Although their average weight was not significantly different between the two groups, 53% of the pups with IGF-1-deficient macrophages weighed <1.2 gram at 24 h of life, in contrast to 38% of the control littermates (Suppl. Fig. 4a, b). Also, at baseline, the relative length of the intestine of pups with IGF-1-deficient macrophages was ~10% shorter (89.8 ± 5.3%) compared with littermate controls of the same sex and weight (Suppl. Fig. 4c, d).

When intestinal sections of 1-day-old DF pups with IGF-1-deficient macrophages were stained by immunofluorescence for Ki-67 (proliferation marker) and endomucin (endothelial cell marker), the number of proliferating endothelial cells in the villous LP was significantly decreased compared to littermate controls (Fig. 5d, e). Also, when assessed by flow cytometry, the percentage of proliferating endothelial cells (Ki-67$^+$CD31$^+$CD45$^-$) was significantly decreased in the intestinal LP of 1-day old pups with IGF-1-deficient macrophages compared to littermate controls, with a marginal decrease at D4 (Fig. 5f, g). However, no significant differences were noted at D0 and D20 (Fig. 5f, g). Also, the percentage of endothelial cells per live LP cells (not shown) and the endothelial cell number relative to the total number of cells isolated per intestine was significantly decreased in D1 pups with IGF-1-deficient macrophages compared to littermate controls (Fig. 5h). When neonatal IGF-1-deficient macrophages were cultured with intestinal endothelial cells, less robust tube-like sprouting structures were seen compared with endothelial cells cultured with IGF-1-sufficient macrophages (Fig. 6a, b). Furthermore, the small intestine of pups with IGF-1-deficient macrophages had lower expression of VEGF compared to the IGF-1-sufficient littermate control group (Fig. 6c, d).

When submitted to an NEC model as previously described[30], we found a significant increase in mortality in pups with IGF-1-deficient macrophages compared to controls, with a median survival of 34 h in the IGF-1-deficient group compared with 44 h in the control group (Fig. 6e). Furthermore, the incidence of severe NEC (score ≥2) was markedly increased in the IGF-1-

deficient group (20/35) compared with the control group (13/46; $\chi^2 = 6.868$; Fig. 6f).

**Exogenous IGF-1 preserves intestinal VEGF/VEGFR2 expression, endothelial cell proliferation, and microvascular density, improves survival, and attenuates tissue injury in experimental NEC.** To examine the role of IGF-1 on intestinal endothelial cell proliferation in vivo, 1-day-old neonatal mice were dam-fed and injected once with PPP (5 mg/kg, i.p.) or vehicle control, or submitted to experimental NEC. Twenty-four hours later, intestines were collected to assess LP endothelial cell proliferation by flow cytometry (Fig. 7a, b). Inhibition of IGF-1 signaling significantly decreased intestinal endothelial proliferation to levels similar to pups submitted to the experimental NEC protocol (Fig. 7a, b). We previously showed that exposure to the NEC protocol significantly decreased intestinal VEGF expression, intestinal endothelial cell proliferation, and intestinal microvascular density in the intestinal villi compared to dam-fed controls[10]. To determine whether these parameters were affected by exogenous IGF-1 during NEC development, neonatal mice were injected with recombinant IGF-1 (50 μg/kg/day) or vehicle control twice daily and submitted to an NEC model for 24 h. Intestinal VEGF and VEGFR2 expression were assessed by western blot analysis, and proliferating intestinal villous endothelial cells and intestinal microvascular density were assessed by immunofluorescence. In pups exposed to the NEC model for 24 h, treatment with IGF-1 preserved intestinal VEGF and VEGFR2 expression, to levels comparable to the control group (Fig. 7c, d). Treatment with IGF-1 but not vehicle control prevented NEC-induced decrease in villous endothelial cell proliferation (Fig. 7e, f) and preserved the intestinal vascular network density (Fig. 7g, h). To determine whether IGF-1 treatment protects the intestine against injury in a neonatal NEC model, 50 μg/kg/day of IGF-1 or vehicle control were injected to pups i.p., twice daily with the first dose given 2 h prior to NEC initiation. In the IGF-1 treatment group, animal survival was significantly improved (median survival of 55 h compared to 41.5 h in control group (Fig. 7i). Furthermore, the incidence of severe histological intestinal injury (grade ≥2) was decreased in the IGF-1 treatment group in comparison to the control group (12/29 vs 19/28, $\chi^2 = 4.03$) (Fig. 7j).

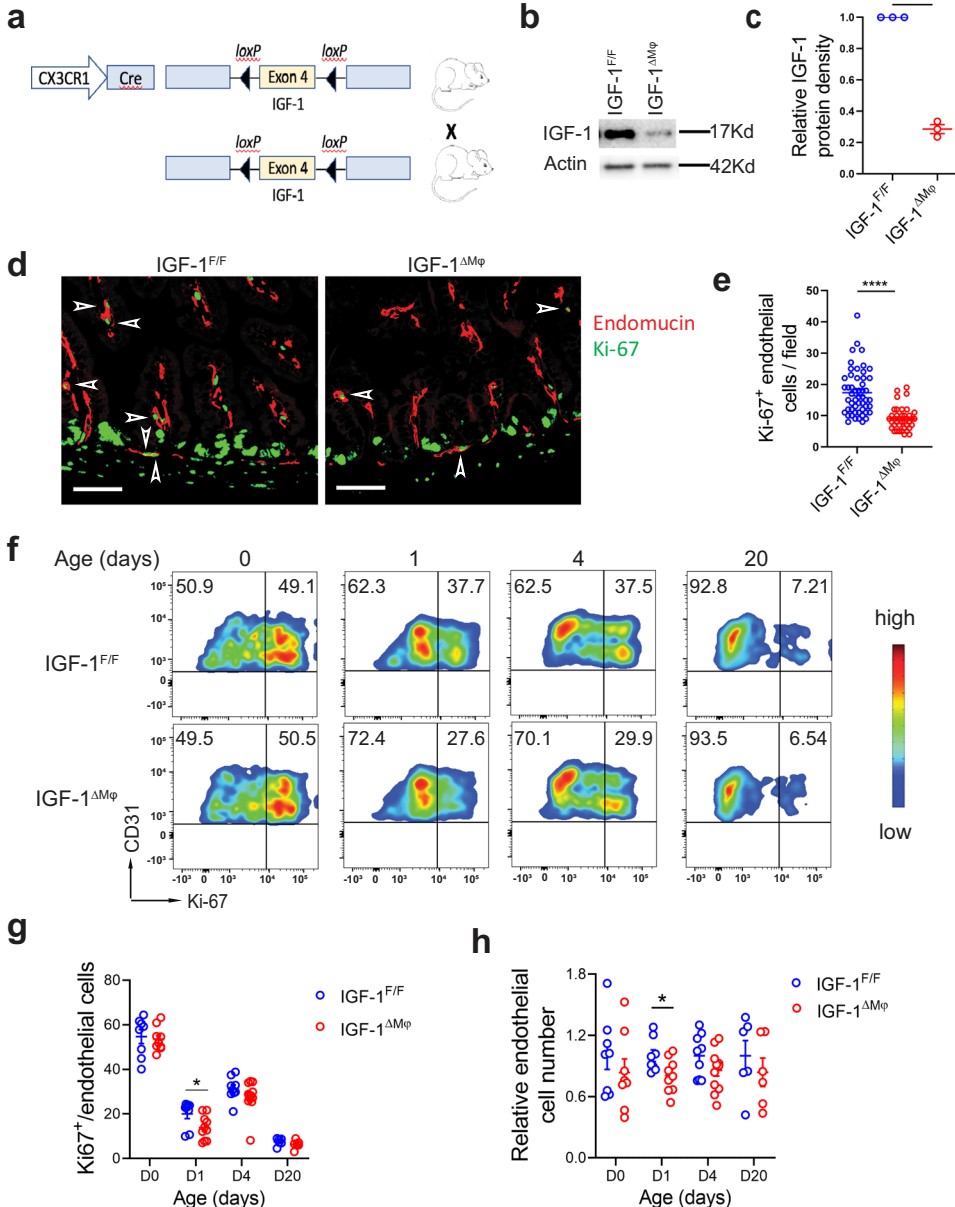

**Fig. 5 Mice with IGF-1-deficient macrophages have decreased intestinal endothelial cell proliferation. a** Breeding scheme for generating mice with IGF-1 sufficient and IGF-1-deficient macrophages. **b**, **c** IGF-1 protein is decreased in isolated CX3CR1+ cells from pups with IGF-deficient macrophages (*Igf-1*ΔMφ) compared with IGF-1-sufficient littermates (*Igf-1*f/f). For each group and in each experiment, intestines from 8 to 10 pups were pooled together to obtain enough cells for western blot analysis. **b** Western blot image of one experiment. **c** The experiment was repeated twice, and band density was quantified (mean ± SEM, *n* = 3 /group). **d**, **e** Tissue sections from 24-hour-old *Igf-1*ΔMφ pups (*n* = 6) or IGF-1-sufficient pups (*n* = 8) were stained for Ki-67 (green) and endomucin (red). **d** Typical image is shown, scale bar: 50 μm. **e** Average proliferating endothelial cells (Ki-67+ endomucin+) per ×20 field in the intestinal villi (see arrows in **d**) were counted from 3 to 4 field images per sample. Each dot indicates the number from one image. **f**–**h** Isolated small intestinal LP cells from D0, D1, D4, D20 IGF-1-sufficient or *Igf-1*ΔMφ pups were gated on single/live CD31+CD45− cells and analyzed for CD31 and Ki-67. **f** Typical flow cytometry images are shown. **g** The graph represents the percentage of proliferating (Ki-67+) endothelial cells at different ages in the intestine of IGF-1-sufficient or *Igf-1*ΔMφ pups (mean ± SEM). **h** Endothelial cell number relative to the isolated total cell number per intestine, which is subsequently normalized to Igf-1f/f group, is shown here (mean ± SEM), *n* = 6–9/group (**g**, **h**, see dots in panels and Source Data file for exact *n* number). *P* values were calculated using two-sided Student's *t* tests (**c**, **e**) or multiple *t* tests (**g**, **h**). *p < 0.05, **p < 0.01, ***p < 0.001, ****p < 0.0001. Source data are provided as a Source Data file.

**In human NEC, endothelial cell proliferation and IGF-1-producing macrophages are decreased in the intestinal villi compared to controls.** To determine whether endothelial cell proliferation is affected in human NEC tissues, intestinal tissue sections from non-necrotic areas of patients with NEC and controls were stained with antibodies against CD31 and Ki-67. Although many proliferating endothelial cells (CD31+Ki-67+

double-positive cells) were present in the villi of control tissues, these were rarer in the villi of NEC tissues (Fig. 8a, b). To confirm the relevance of our mouse findings to human NEC, we examined whether macrophages produce IGF-1 in the human neonatal intestine. To do so, IGF-1 mRNA was assessed by in situ hybridization in a human intestinal specimen obtained from an infant undergoing surgery for reanastomosis (control), and IGF-1

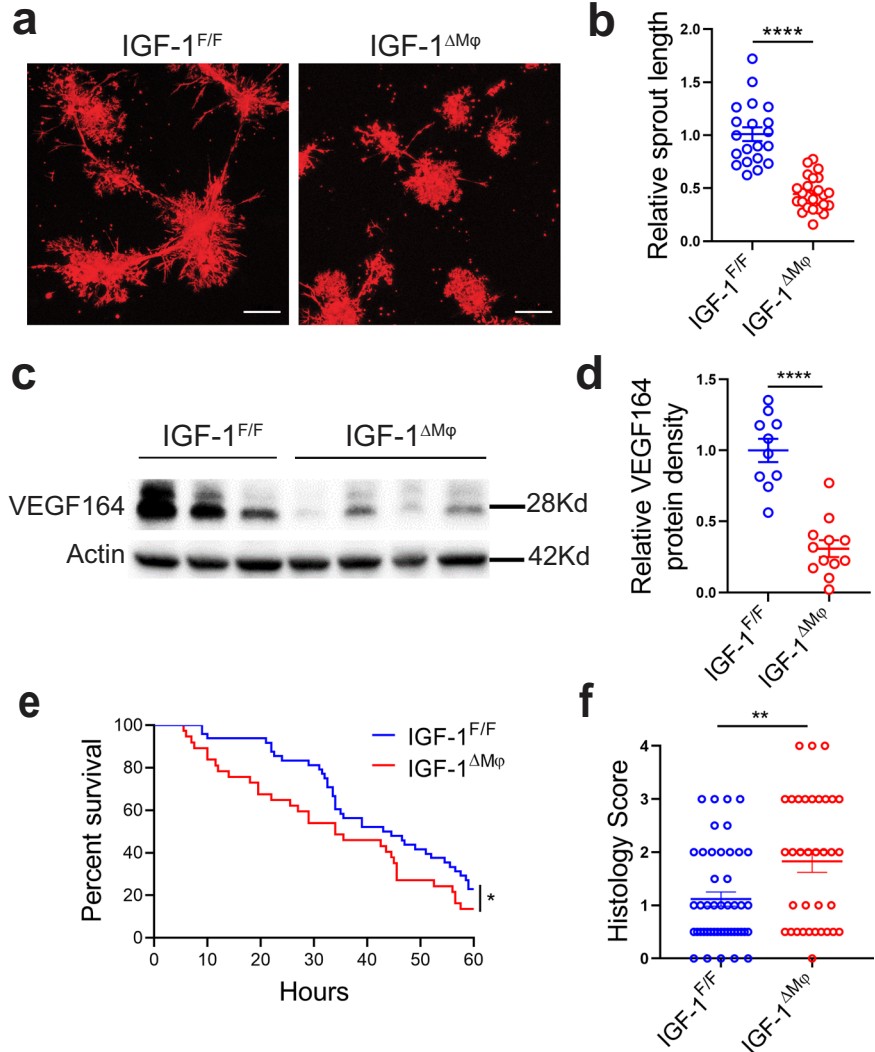

**Fig. 6 IGF-1-deficient macrophages have decreased proangiogenic properties and *Igf-1*$^{\Delta M\varphi}$ mice are more susceptible to experimental NEC. a**, **b** $4 \times 10^4$ intestinal endothelial cells from mT/mG neonatal pups were co-cultured for 4 days on Matrigel with $2 \times 10^4$ CX3CR1$^+$ neonatal intestinal macrophages obtained from IGF-1-sufficient ($n = 6$) or *Igf-1*$^{\Delta M\varphi}$ ($n = 6$) pups. **a** Typical sprouting images are shown. **b** Sprout length was measured, each data point indicates the length of a single sprout (mean ± SEM). **c**, **d** Intestinal tissue VEGF expression was assessed at D1 (24 h of life) by western blot analysis and relative band density to actin was quantified (mean ± SEM), $n = 10$ and 12 IGF-1-sufficient or *Igf-1*$^{\Delta M\varphi}$ groups, respectively. *P* values were calculated using two-sided Student's *t* tests (**b**, **d**). **e**, **f** IGF-1-sufficient or *Igf-1*$^{\Delta M\varphi}$ pups were submitted to the neonatal NEC model. **e** Survival curves are shown, $n = 48$ and 37 in IGF-1-sufficient or *Igf-1*$^{\Delta M\varphi}$ groups, respectively, *P* values were calculated using Gehan–Breslow–Wilcoxon test. **f** Intestinal histological injury scores are shown, $n = 46$ and 35 in IGF-1-sufficient or *Igf-1*$^{\Delta M\varphi}$ groups, respectively, *P* values were calculated using Chi-square analysis. All plots are the results of three separated experiments combined. *$p < 0.05$, **$p < 0.01$, ****$p < 0.0001$. Source data are provided as a Source Data file.

mRNA was detected in macrophages (CX3CR1$^+$ cells) (Fig. 8c). Second, sections from non-necrotic areas of patients with NEC and controls were stained with antibodies against IGF-1 and CX3CR1 and examined by immunofluorescence staining. We found a marked decrease in the number of IGF-1-producing intestinal macrophages (CX3CR1$^+$ cells) in human NEC tissues compared to controls (Fig. 8d, e).

## Discussion

NEC is a highly prevalent intestinal disease of premature infants associated with high mortality[1]. Specific therapy is not currently available because its causes are not well understood. Low levels of serum IGF-1 have been found to predispose premature neonates to NEC[18] and recent evidence suggests that defective intestinal microvascular development significantly contributes to NEC[6]. We show here that, in the neonatal intestine at birth, macrophages are closely juxtaposed to intestinal endothelial cells of the villi, and,

upon sensing endothelial cells, these macrophages produced increased levels of IGF-1 and promote intestinal VEGF expression, vascular endothelial cell proliferation, and mucosal microvasculature development via an IGF-1-dependent mechanism. Neonatal mice with IGF-1-deficient macrophages are more susceptible to NEC. Exogenous IGF-1 treatment preserves intestinal tissue VEGF protein expression and endothelial cell proliferation and protects neonatal mice against experimental NEC. Our study also reveals that, in humans, macrophages express IGF-1 in the neonatal intestine and that, in human NEC tissues, the number of IGF-1$^+$ macrophages and endothelial cell proliferation is significantly decreased. Together, these findings suggest that, in the neonatal intestine, macrophage-dependent IGF-1 signaling is critical for normal mucosal microvascular development and its deficiency predisposes to NEC.

Embryonically derived macrophages are the main myeloid cells of the small intestine in neonatal mice[23]. These cells get diluted

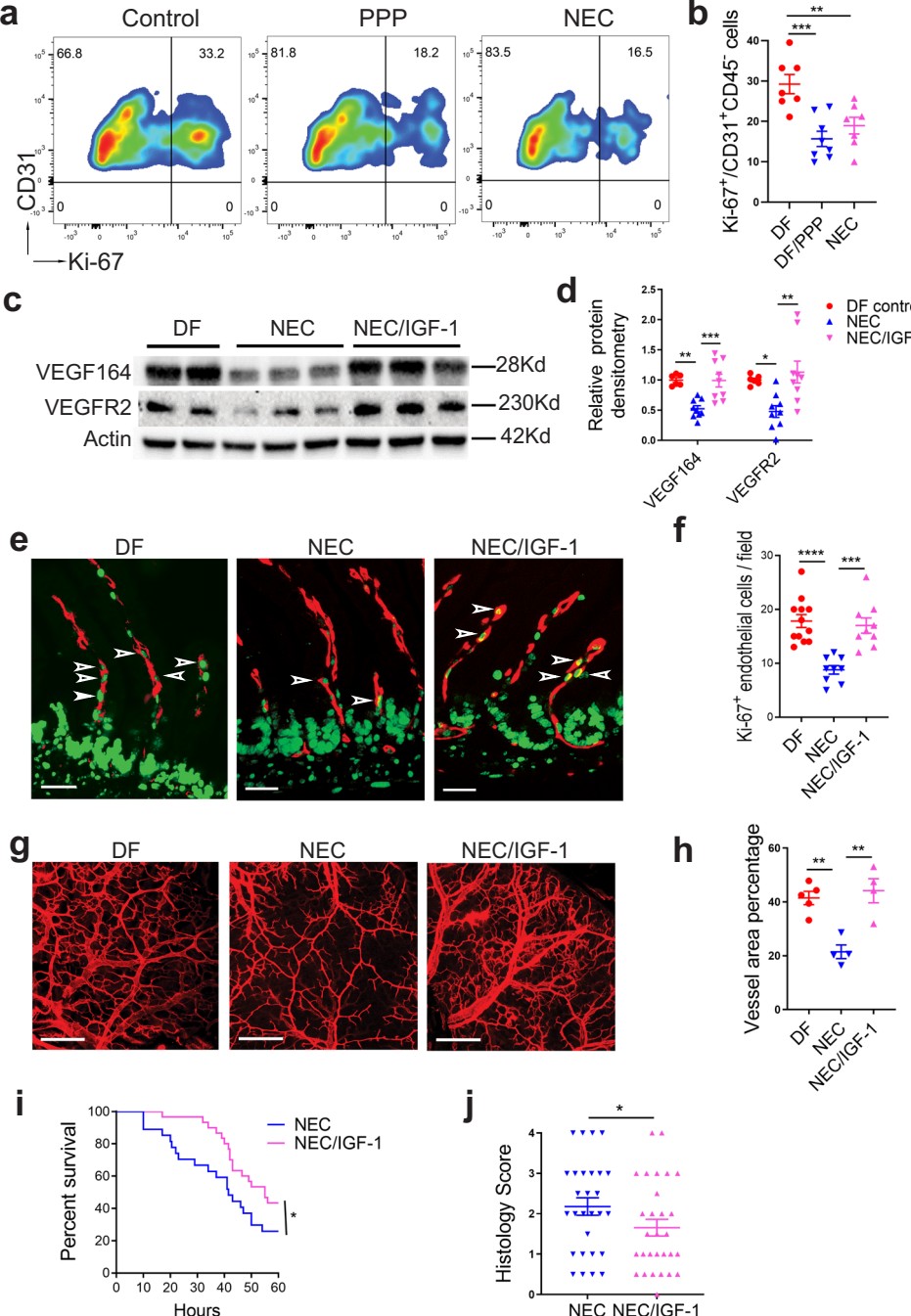

**Fig. 7 Blocking IGF-1 signaling decreases intestinal endothelial cell proliferation in vivo; exogenous IGF-1 enhances VEGF protein expression, intestinal endothelial cell proliferation and preserves intestinal microvascular density during experimental NEC. a, b** 24-hour-old wild-type C57BL/6 pups were treated with the IGF-1R inhibitor PPP ($n = 8$) or vehicle control ($n = 7$) or submitted to the NEC protocol for 24 h ($n = 7$). Intestinal LP cells were isolated and the percentage of proliferating endothelial cells (Ki-67⁺CD31⁺CD45⁻/CD31⁺CD45⁻) was determined by flow cytometry. **a** Typical FACS plots are shown. **b** Graph represents the results of three experiments combined (mean ± SEM). **c–h** 1-day-old neonatal mice were injected with IGF-1 (25 μg/kg, i.p., twice) or vehicle control, with a first dose 2 h prior to NEC initiation and a second dose 12 h later (Dam-fed littermates as controls) then submitted to experimental NEC for 24 h (**c–f**) or 48 h (**g**, **h**). **c, d** Intestinal tissues were assessed for VEGF and VEGFR2 proteins by western blot (**c**) and band densitometry is shown (**d**, mean ± SEM), $n = 6$–9/group (see panel or Source Data file for exact $n$ number). **e, f** Tissue sections were stained with Ab against Ki-67 and endomucin. **e** Typical immunofluorescence images. **f** Graph represents average proliferating endothelial cells (Ki-67⁺ endomucin⁺ cells, arrows in **e**) counted per ×20 field (mean ± SEM, three fields were taken from each sample), $n = 9$–12/group (see Source Data file for exact $n$ number). **g, h** pups were perfused intracardially with WGA-Alexa Fluor 647. Intestinal submucosal vascular networks were imaged, and vascular density was assessed using Photoshop (×10), $n = 4$–5/group (see dots in panel). *P* values were calculated using one-way ANOVA followed by Turkey–Kramer multiple-comparison test (**b**, **d**, **f**, and **h**). **i, j** Neonatal mice was submitted to the NEC model and treated with IGF-1 or control twice daily (see method section). **i** 60-hour-survival curve is shown ($n = 28$ in NEC and 30 in NEC/IGF-1 group), *P* value were calculated using Gehan–Breslow–Wilcoxon test. **j** Intestinal injury histological scores ($n = 28$ in NEC and 29 in NEC-IGF-1 group), *P* value were calculated using Chi-square test. *$p < 0.05$, **$p < 0.01$, ***$p < 0.001$, ****$p < 0.0001$. Source data are provided as a Source Data file.

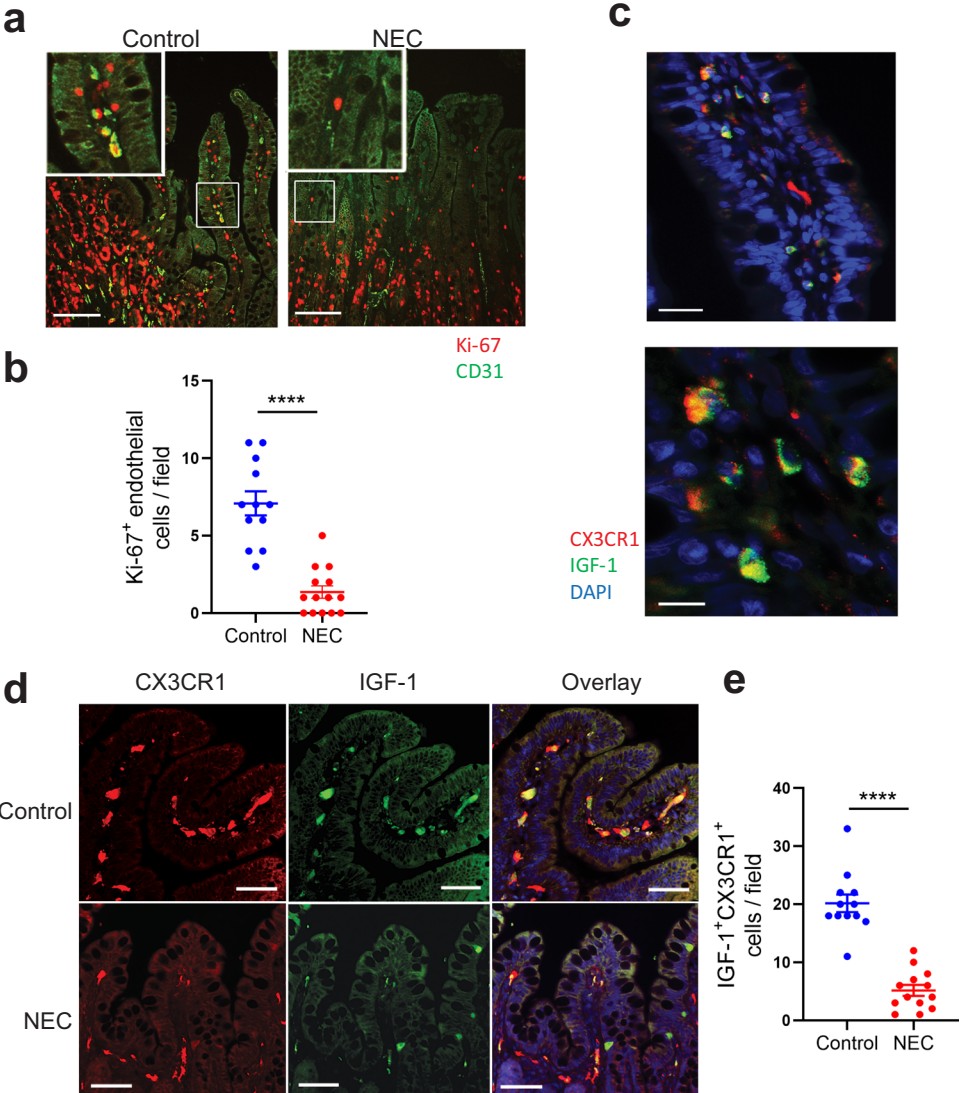

**Fig. 8 Endothelial cell proliferation is diminished and the number of IGF-1-producing macrophages is decreased in the intestine of human neonates with NEC compared to controls. a, b** Sections of human NEC ($n = 4$) and control ($n = 5$) intestinal tissues were stained for CD31 (green) and Ki-67 (red). 2–3 images per sample from non-necrotic area were taken at ×20 and proliferating endothelial (CD31+Ki-67+) cells were counted. **a** Typical images are shown here. **b** Number of proliferating endothelial cells (represented by each dot) per ×20 image (mean ± SEM), scale bar = 100 μm. **c** Images of control human neonatal intestinal tissue sections subjected to in situ hybridization using a probe for IGF-1 (green) and counterstained for macrophages using anti-CX3CR1 antibodies (red); bar = 25 (top) and 10 μm (bottom). **d, e** NEC and control tissue sections were stained by immunofluorescence for IGF-1 (green) and the macrophage marker CX3CR1 (red). **d** Typical images are shown. Overlay image includes DAPI, scale bar = 100 μm. **e** Graph represents the number of IGF-1+ CX3CR1+ cells per field (mean ± SEM, 3–4 ×20 fields of non-necrotic area were taken from each sample; $n = 4$ patients/group). P values were calculated using two-sided Student's t tests (**b** and **e**). ****$p < 0.0001$. Source data are provided as a Source Data file.

out by monocyte-derived macrophages during postnatal development in mice[23]. Here, we show that neonatal intestinal macrophages promote intestinal endothelial cell proliferation and mucosal microvascular development. Also, although the mechanism remains to be explored, our in vitro data suggest that macrophages sense endothelial cells to produce increased levels of IGF-1. Macrophages have been shown to play a role as accessory cells to promote angiogenesis in the heart. In the injured heart, neonatal mouse embryonic-derived macrophages but not adult macrophages promote cardiac recovery and efficiently stimulate endothelial cell tube formation in vitro[31]. In the embryo, yolk sac-derived embryonic macrophages promote coronary development via IGF-1[24]. Our data are the first to suggest a role for macrophage-derived IGF-1 in promoting neonatal intestinal microvascular development.

IGF-1 has a dual-mode action, being both an endocrine and paracrine growth factor. Circulating IGF-1 is mainly produced by the liver and acts as the primary mediator of growth hormone (GH)-dependent growth, as an important mitogenic factor regulating growth, nutrient metabolism, reproduction, and aging[32,33], whereas local IGF-1 is produced by peripheral tissues acting as a paracrine/autocrine factor for local tissue growth, probably working together with circulating IGF-1[33]. Our data show that both plasma and intestinal tissue IGF-1 are developmentally regulated, being very low in the first few days of life in mice. We found that both serum and intestinal IGF-1 are decreased by exposure to the experimental NEC protocol. This observation is congruous with previous studies showing that IGF-1 plasma levels are decreased during inflammation in animal models[34,35] and in humans[34,36]. The decreased plasma level may

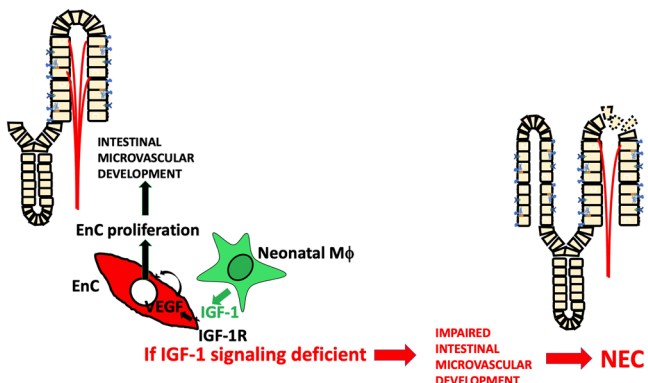

**Fig. 9 IGF-1 released from embryonically derived macrophages binds to IGF-1R on endothelial cells to activate VEGF expression, leading to endothelial cell proliferation, promoting intestinal microvascular development. Deficient macrophage-derived IGF-1 signaling impairs intestinal microvascular development, thus increasing intestinal susceptibility to NEC.** EnC endothelial cell(s), IGF-1 insulin-like growth factor, IGF-1R IGF-1 receptor.

be caused by proteolytic degradation of IGF-1 binding protein-3 resulting in an increased clearance of IGF-1, as previously shown in a model of juvenile arthritis induced by IL-6 overexpression[37]. Our human data suggest that the decrease in intestinal IGF-1 may be due to the loss of IGF-1-producing macrophages. As intestinal vascular endothelial cell proliferation is mostly impacted during the first few days of life in macrophage-IGF-1-deficient mice, we speculate that, the paracrine effect of macrophage-derived IGF-1 on endothelial cells may be especially critical during the early neonatal period when plasma levels of IGF-1 are particularly low. Interestingly both IGF-1 and IGF-1R, after a sharp decrease immediately after birth, are highly expressed in the mouse intestine by the end of the first week of life. Both IGF-1 and IGF-1R significantly decrease between 7 and 21 days of age, which corresponds to the weaning period. As breast milk contains a significant amount of IGF-1[16], it may play an important role in promoting IGF-1 signaling in the intestine during this critical period and further studies are needed in this regard.

Our study identifies "defective mucosal intestinal microvascular development" as a novel mechanism by which decreased IGF-1 signaling leads to intestinal injury and NEC. There have only been a few studies on the role of IGF-1 on intestinal microvasculature. One such study showed an association between low IGF-1 mRNA levels and decreased microvascular density in colorectal cancer tissues[38], suggesting a role for IGF-1 in intestinal angiogenesis. Primary human microvascular endothelial cells of the skin have been shown to express IGF-1R, which mediated tip cell formation and sprouting[39]. In a denuding arterial injury model, endothelial IGF-1R has been shown to mediate endothelium regeneration[40]. Our data show that intestinal endothelial cell proliferation and mucosal microvascular development in neonatal mice is under the regulation of IGF-1, and that embryonic macrophages in the intestine promote microvascular development via IGF-1. However, our study could not differentiate an effect arising from the macrophages derived from the LP vs. the muscularis mucosa. In our study, exogenous IGF-1 (at the dose of 50 μg/kg/d) attenuated tissue injury in mice exposed to the NEC protocol while preserving intestinal endothelial cell proliferation, microvasculature density, and VEGF/VEGFR2 expression. Besides its protective effect on the microvasculature, IGF-1 has also been shown to protect intestinal epithelial cells from oxidative stress-induced apoptosis[41]. Also, enterally administered IGF-1 has been shown to down-regulate TLR4 signaling, reducing the

inflammatory response and protecting against intestinal injury in a rat NEC model[42]. Together, this suggests that the protective effect of IGF-1 in the neonatal intestine occurs at several levels.

In summary, this study provides new evidence to support the paracrine role of macrophage-derived IGF-1 on promoting neonatal intestinal microvasculature development (Fig. 9). Furthermore, this study provides supporting evidence of a protective effect of exogenous IGF-1 on NEC development by preserving intestinal microvascular development. Further investigations are needed to define interventions that can preserve the paracrine role of embryonically derived macrophages in premature neonates to prevent NEC development.

## Methods

**Reagents**. IGF-1 ELISA kit (MG100) was purchased from R&D (Minneapolis, MN). Recombinant murine IGF-1(250-19) was obtained from PeproTech (Rocky Hill, NJ). Anti-CD31 antibodies (Ab) (ab28364, for immunofluorescence stain), anti-F4/80 (ab6640, for whole-mount staining) were purchased from Abcam (Cambridge, UK). Anti-CD31 Ab (102419, for FACS analysis), Biotin-labeled anti-mouse CX3CR1 Ab for cell isolation (149018), Biotin anti-CX3CR1 Ab for human tissue immunofluorescence staining (824004), Ly6C Ab for FACS (128037) and F4/80 APC for FACS (123115) were obtained from Biolegend (San Diego, CA). CD11c APC Ab (R700) and NK1.1 Ab (BV786) for FACS were purchased from BD Biosciences. Biotin anti-mouse CD11b Ab (553309) used for cell isolation was purchased from BD Biosciences (Franklin Lakes, NJ). Anti-IGF-1 Ab (sc-518040), anti-IGF-1R Ab (sc-462) and their isotype controls (for FACS), anti-VEGF-A Ab (sc-7269), and anti-IGF-1Rα Ab (sc-463) for western blotting were obtained from Santa Cruz Biotechnology (Santa Cruz, CA). Anti-IGF-1 Ab for western blotting (NBP2-16929) was purchased from Novus Biologicals (Littleton, CO). Anti-mouse Tie-2 (13-5987-82), Anti-CD45 (17-0451-82), anti-endomucin (50-5851-80), anti-Ki-67 (MA5-14520, for immunofluorescence stain), anti-Ki-67 (for FACS), Streptavidin Alexa Fluor 594 conjugate (S11227), goat anti-rabbit (A11037, A11034), goat anti-rat (A11007) donkey anti-goat (A21432), donkey anti-rat (A21208) Ab, FocusClear (FC-101), Alexa Fluor 647-conjugated wheat germ agglutinin, and a live/dead cell kit (L23105), collagenase IV (17104019) were obtained from Fisher Scientific (Waltham, MA). Anti-VEGFR2 (2479, for western blot) Ab were purchased from Cell Signaling Technology (Danvers, MA). Picropodophyllin (PPP) was obtained from Tocris Bioscience (Bristol, UK). Endothelial cell culture media (LL-0004 and LL-0005) was obtained from Lifeline Cell Technology (Frederick, MD). Collagenase VIII (C2139) was purchased from Sigma-Aldrich (St. Louis, MO). FcR blocking reagent (130-092-575), anti-biotin microbeads (130-090-485), mouse CD45 (130-052-301), and CD31 microbeads (130-097-418) were purchased from Miltenyi Biotec (Bergisch Gladbach, Germany). Growth Factor Reduced Matrigel (354230) was obtained from Corning Incorporated (Corning, NY). Anti-VEGFR2 (AF647, for whole-mount staining) was obtained from R&D system (Minneapolis, MN).

**Human intestinal tissue collection**. The study was approved by the Lurie Children's Hospital of Chicago Institutional Review Board. All legal guardians of the participants signed informed consent forms. Intestinal surgical specimens of NEC patients and controls were fixed in 10% formalin and were paraffin-embedded. Patients 'characteristics are described in Table 1.

**Breeding strategy**. Breeders were housed at the ratio of one male to a maximum of three females and overnight breeding strategy was used to generate pups for experiments. Breeder chow was fed to breeders. To be able to identify littermates for each experiment, pregnant dams were housed individually 3 days prior to delivery. Age between 8 weeks and 6 months of female breeders were typically used for breeding. To minimize possible confounding effects of birth weight or sex on experiment results, special care was applied to use groups of littermates with equal weight and sex distribution. To determine the sex of neonatal pups, newborn mice male gender was identified by the pigmented spot on the scrotum[43].

**Animal experiments**. Commercially available animals including C57BL/6, Gt(ROSA)26Sor$^{tm4(ACTB-tdTomato,-EGFP)Luo}$/J (mT/mG), B6.129P2(Cg)-Cx3cr1$^{tm1Litt}$/J (Cx3cr1-GFP), B6J.B6N(Cg)-Cx3cr1$^{tm1.1(cre)Jung}$/J (Cx3cr1-Cre), B6.129(FVB)-Igf1$^{tm1Dlr}$/J (Igf1$^{flox}$) were purchased from Jackson Laboratory (Bar Harbor, ME). All animal breeding, maintenance, and procedures were approved by the Institutional Animal Care and Use Committee and animal use followed the regulation of the Center for Comparative Medicine of Northwestern University.

Pups <24-hour-old were submitted to the NEC model as previously described[30] which includes: (1) initial orogastric inoculation with a standardized adult mouse commensal bacteria preparation ($10^8$ colony-forming units) and LPS (5 mg/kg) to perturb the normal intestinal colonization process; (2) gavage with formula every 3 h (Esbilac, 200 ml kg$^{-1}$ day$^{-1}$); and (3) exposure to brief episodes of hypoxia (60 s in 100% $N_2$) followed immediately by cold stress (10 min at 4 °C) twice daily.

**Table 1 Human intestinal tissue patient characteristics.**

| Diagnosis | Tissue section | Gestational age (weeks) | Birth weight (grams) | Gender | Age at surgery | Maternal betamethasone |
|---|---|---|---|---|---|---|
| Multiple intestinal atresia | Ileum | 36 2/7 | 2795 | Male | 0 day | No |
| Ileal stricture removal | Distal ileum | 26 | 1145 | Male | 14 weeks | (not documented) |
| Ileal stricture removal | Ileum | 23 3/7 | 480 | Male | 8 ½ weeks | 1 dose |
| Colonic stricture removal | Colon | 29 | 1260 | Male | 9 weeks | No |
| Gastroschisis | Ileum | 38 | 3010 | Male | 9 weeks | No |
| NEC | Distal ileum | 32 3/7 | 1655 | Male | 9 ½ weeks | Full course (at 25 weeks of gestational age) |
| NEC | Distal ileum | 33 | 1800 | Male | 12 days | Full course |
| NEC | Distal ileum | 32 5/7 | 1690 | Male | 7 weeks | Full course |
| NEC | Jejunum | 24 | 569 | Female | 3 months | (not documented) |

This protocol induces intestinal injuries ranging from epithelial injury to transmural necrosis resembling human NEC[30] which typically develops after 36 h and has been widely used[44–47] to study NEC pathogenesis. In some experiments, <24-hour-old pups were submitted to the NEC protocol and treated (i.p.) either with 50 μg/kg/d of recombinant murine IGF-1 diluted in 0.9% saline/1% sodium carboxymethycellulose or vehicle control, which was administrated twice daily with the first dose given 2 h prior to NEC protocol initiation. Pups were closely monitored and euthanized when showing signs of distress. Whole intestinal tissues were collected. Hematoxylin and eosin-stained tissue sections were evaluated and scored by two investigators unaware of the group assignment. Severe NEC was defined as a histological score ≥2. Pups found dead while not under direct observation were excluded from histological analysis. Alternatively, small intestinal tissues were collected 6–48 h following NEC initiation for experiments studying the mechanism leading to NEC.

**Western blotting.** Small intestinal tissue lysates were obtained by tissue homogenization in lysis buffer (10 mM Tris·HCl pH 7.6, 150 mM NaCl, 5 mM EDTA, 1 mM PMSF, 1 mM DTT, 0.25% Nonidet P-40, and protease inhibitor). The protein concentration of the tissue lysates was determined with the Bradford method and 20–50 μg of protein were separated on 4–15% pre-cast gradient sodium dodecyl sulfate–polyacrylamide gel electrophoresis gel. Proteins were transferred onto nitrocellulose or polyvinylidene fluoride membranes. Membranes were blocked with 5% milk/Tween 20 for 60 min, incubated with primary antibody at 4 °C overnight, then with horseradish peroxidase (HRP)-conjugated secondary antibodies for 1 h at room temperature. Target proteins were detected using the standard Pierce enhanced chemiluminescence method. β-actin was probed on the same membranes to serve as an internal control. Alternatively, epithelial cell fraction, isolated macrophages and endothelial cells (see below "cell isolation" section) were counted and washed in PBS. Cell pellet was directly lysed in 2× sample buffer, boiled for 5–10 min and 100,000–200,000 cells-generated lysate was loaded per lane for western blotting analysis as above.

**Immunofluorescence staining.** Five-micrometer-thick tissue sections were prepared from formalin-fixed and paraffin-embedded whole intestine. Antigen retrieval was performed by boiling tissue sections in pH 8.5 EDTA buffer for 20 min. After being blocked with 10% normal goat serum for 1 h at room temperature, sections were incubated with primary Ab at 4 °C overnight. Fluorescent conjugated secondary Ab was applied to the section and incubated at room temperature for 1 h. For human tissue, sections were stained with anti-IGF-1 AF488 Ab and biotin-conjugated anti-CX3CR1 Ab followed by AF 594 conjugated streptavidin staining. Images were captured on a Leica DMR-HC upright microscope after DAPI-included mounting media was added. Alternatively, for whole-mount intestine immunostaining, pups were intracardially perfused with 4% paraformaldehyde solution, processed precisely as previously described[48]. One centimeter length of tissue, which distances 2 centimeters from the stomach, was stained with anti-VEGFR2 and anti-F4/80 antibodies overnight followed by donkey anti-goat AF555 and donkey anti-rat AF488 antibodies overnight. Tissues were then cleared with FocusClear for 20-30 min and confocal images were taken on a Leica 880 microscope[48]. Macrophage cell size, perimeter, and circularity were analyzed using photoshop software.

**In situ hybridization.** Total RNA was extracted from human intestinal tissue and primers for IGF-1 that included a T7 sequence on the reverse primer (F-TGTAT TGCGCACCCCTCAA, R-TAATACGACTCACTATAGGGTCCAGCAGCCAAG ATTCAGA) were utilized to create template cDNA capable of subsequent in vitro reverse transcription. This reaction was used to generate a 510 bp DIG-labeled probe (Roche) that recognizes human IGF-1 mRNA. In situ hybridization was performed as previously described[49]. Briefly paraffin-embedded tissues sections

were deparaffinized in xylene and rehydrated in a series of ethanol washes. Sections were digested with proteinase K (Roche) for 20 min at 37 °C, washed then blocked with hybridization buffer for 1 h. Sections were hybridized overnight with the DIG-labeled probe diluted in hybridization buffer at 60 °C in a formamide humidified chamber. Slides were washed three times for 20 min each at 62 °C. Finally, slides were incubated with an anti-DIG HRP-conjugated antibody (Roche), followed by TSA Alexa 488 (Thermo Fisher) according to the manufacturer's instructions. Slides were mounted using Prolong Gold with DAPI (Thermo Fisher).

**Cell Isolation.** Small intestinal tissues were collected from mice at different ages and mesenteric tissues were removed completely under a dissection microscope. Intestines were cut open and rinsed in PBS to remove feces. Tissue was then cut into small pieces and shaken in dissociation buffer (PBS: 2% FBS, 1.0 mM DTT, 5 mM EDTA, 15 mM HEPES) for 30 min at 37 °C to release epithelial cell fraction. The rest of epithelial-free LP tissue was digested at 37 °C for 45 min in DMEM/ 1 mg/ml collagenase IV/5 mM CaCl₂/15 mM HEPES solution. Digested cell solution was filtered through 40 μm strainer, washed in PBS/2% FBS/2 mM EDTA to be ready for magnetic cell isolation using indirect or direct microbeads staining. Myeloid cells were isolated using streptavidin microbeads selection after cells were stained with biotin-labeled CD11b antibody, and macrophages were purified using a similar strategy after biotin-labeled CX3CR1 antibody staining. CD31$^+$ endothelial cells were isolated using anti-CD31 microbeads after CD45$^+$cells were depleted by using anti-CD45 microbeads[50].

**LP cell flow cytometry.** Isolated total small intestinal LP cells were prepared as above (see "cell isolation" except Collagenase VIII was used instead of Collagenase IV) washed in PBS twice before being stained with live/dead marker. Cells were then FcR-blocked and stained with anti-CD31, CD45 antibodies, then with anti-Ki-67 antibody after fixation and permeabilization using FOXP3 staining kit. To determine the total endothelial cell number per intestine in $Igf\text{-}1^{f/f}$ $Cx3cr1\text{-}cre^{+/-}$ and $Igf\text{-}1^{f/f}$ littermates, the flow data was normalized to the total LP cell number per intestine counted by trypan blue exclusion. Markers including Ly6C, GFP (CX3CR1), CD11c, NK1.1, F4/80 were also used to characterize CX3CR1$^+$ cells in the neonatal intestine. Data acquisition was performed on an LSRFortessa cell analyzer and analyzed using FlowJo software (version 10.7.1) by gating live and singlet cells.

**Endothelial cell/macrophage co-culture experiments.** Intestinal endothelial cells from mT/mG pups were purified as previously described[50]. Myeloid cells were purified from either $Cx3cr1$-GFP or wild-type (WT) pup intestinal tissues. All cell isolations were performed by pooling 8–15 pup intestines in each group to yield enough cells for experiments. In all, $4 \times 10^4$ endothelial cells and $2 \times 10^4$ myeloid cells or macrophages were cultured on Matrigel separately or together, in endothelial culture media without or with IGF-1 (100 ng/ml) and/or PPP (500 nM). Cell sprouting images were taken by a Zeiss 880 Confocal Laser Scanning Microscope at 10x magnification. The length of the sprouts from each image was measured using photoshop. Alternatively, isolated macrophages and endothelial cells were cultured in vitro either separately or together at a 1:2 ratio in triplicates in 24-well plates. Cells were collected after 48 h of culture and stained with indicated markers or isotype control antibodies for flow cytometric analysis after gating on live singlet cells. IGF-1 was stained after cell fixation and permeabilization using eBioscience™ Intracellular Fixation & Permeabilization Buffer (88-8824, Thermo Fisher) post cell surface marker staining. Endothelial cells were identified as Tie-2$^+$CX3CR1$^-$ population since the CD31 marker could not be used, being no longer detectable after cells have been cultured in vitro and retrieved.

**Intestinal microvasculature imaging.** Pups were anesthetized and Alexa Fluor 647-conjugated-wheat germ agglutinin (500 μl of 40 μg/ml) was infused intracardially as previously described[50]. Whole intestinal tissues were collected

immediately after perfusion and fixed in formalin. Intestinal microvascular network images were captured by a Zeiss 510 META confocal Laser Scanning Microscope and vessel areas were analyzed as previously described[50].

**Statistics**. The statistical software GraphPad Prism (version 8.1.2) was used for statistical analysis. Animal survival data were analyzed by the Gehan–Breslow–Wilcoxon test. Non-parametric chi-square ($\chi^2$) test was used to compare the incidence of severe NEC (Grade $\geq 2$) between two groups. For other data, two-sided Student's $t$ test was used for comparison between two groups, and a one-way analysis of variance (ANOVA) was used for comparison of three or more groups. For ANOVA test, a correction for multiple comparisons was applied for pairwise comparison after ANOVA was performed. All variables were tested for normality by Shapiro–Wilk's test and plot. Data were normalized as needed. Results are expressed as means ± standard error of mean. Differences were considered statistically significant when $P \leq 0.05$, and the level of significance are assigned as $*p < 0.05$, $**p < 0.01$, $***p < 0.001$, and $****p < 0.0001$.

**Reporting summary**. Further information on research design is available in the Nature Research Reporting Summary linked to this article.

## Data availability

No novel data sets were generated in this study. Data that support the findings of this study are available from the corresponding author upon reasonable request. Gt(ROSA) 26Sor[tm4(ACTB-tdTomato,-EGFP)Luo]/J (mT/mG), B6.129P2(Cg)-Cx3cr1[tm1Litt]/J (Cx3cr1-GFP), B6J.B6N(Cg)-Cx3cr1[tm1.1(cre)Jung]/J (Cx3cr1-Cre), B6.129(FVB)-Igf1[tm1Dlr]/J (Igf1[flox]) were commercially available as described in the Method section. Source data for the graphs and charts in the figures and uncropped blots are provided in Supplementary Data 1.

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

## Acknowledgements

We would like to thank Marissa Docter for her technical assistance in conducting the NEC model. This work was funded by the National Institute of Health Grants R01 DK116568 (I.G. De Plaen) and R01 GM117628, DK129960 (X. Tan), by Takeda Pharmaceuticals (I.G. De Plaen) and the Stanley Manne Children's Research Institute of the Ann & Robert H. Lurie Children's Hospital of Chicago (I.G. De Plaen).

## Author contributions

X.Y. performed the in vivo and in vitro experiments. L.M. performed in situ hybridization study and contributed to the in vivo PPP study. X.Y. and I.D.P. were involved in the overall design of experiments and interpretation of results. X.Y. and I.D.P. performed histological evaluation of tissue samples. X.D.T. and Y.Y.Z. provided intellectual input for the project. X.Y. and I.D.P. wrote the manuscript with input from all authors. I.D.P. conceived and orchestrated the project.

## Competing interests

I.D.P. received a grant from Shire Human Genetic Therapies, Inc., a member of the Takeda group of companies, for testing rhIGF-1/BP3 in experimental NEC. X.Y., E.M, Y.Y.Z., and X.D.T. have no competing interests to declare.
