## [Peer Review File · Communications Biology]

Reviewers' comments:

Reviewer #1 (Remarks to the Author):

This is a generally well-written and interesting article on the impact of macrophage-derived IGF-1 on the microvascular development in the neonatal intestine and a thereof resulting protective effect against necrotizing enterocolitis (NEC). It links reduced systemic IGF-1 levels but also impaired local IGF-1 production to the possible development of NEC. A positive effect of intestinal epithelia cell (IEC)-derived IGF1 on the intestinal homeostasis and barrier function and its protective effect against NEC is actually known (PMID: 29783855; PMID: 29131241), wherefore the story is basically not novel. However, to my knowledge, here the authors elucidate for the first time more focused the local effect of macrophage- and endothelia-derived IGF-1 for a healthy intestinal development and NEC protection.

By using a range of different methods, the authors are able to present diverse experimental murine data and add some human data to translate their murine results. However, at some points the manuscript lacks some methodological details that – from my point of view – are important for a better understanding of the drawn conclusions.

The authors may be able to improve the manuscript by addressing the following points:

1. In general, I would recommend improving the phenotype description of the cells of interest.

E.g., I wonder whether using CX3CR1 as single marker for the isolation/identification of lamina propria macrophages is sufficient, as CX3CR1 also is expressed in monocytes, NK cells and dendritic cells (PMID: 10805752; PMID: 15653504). At least for flow cytometry analyses it would be better to use a combination of more specific markers (e.g., F4/80+, CD11b+, Ly6G- etc.) or show functionally that the majority of cells purified from murine intestinal tissue as described in the methods section indeed are macrophages and for instance not dendritic cells.

In addition to this, I recommend to provide a full gating strategy for all FACS analyses (in the supplement) including a detailed description of how the different cell types were identified. Data presented for instance in Fig. 3e otherwise might be misleading as it looks like there are no Tie-2 negative cells at all present in the co-culture.

2. A description is missing of how epithelial cells, macrophages and endothelial cells were isolated/purified from murine tissue to generate protein lysates for the western blot analysis shown in Fig. 2d. In the methods section on western blotting, only the use of small intestine lysates is mentioned. For better understanding and conclusiveness, I suggest preparing an extra paragraph in the methods section to describe cell isolation procedures from murine intestinal tissue for respective assays used in this work.

3. The benefit of Fig. 3d is unclear and not explained in the text (no novel information for the readership). I would recommend omitting it. In addition, the results shown in Fig. 5h do not seem to be very meaningful considering the results regarding intestine length shown in Suppl. Fig. 1c. You would expect a lower cell number if the intestine is significantly shorter in pups with IGF-1-deficient macrophages. I would recommend using relative values (i.e. relative to the isolated overall cell number) for comparison of the two groups.

4. With their work, authors contribute to elucidate a potential mechanism by which IGF-1 protects from NEC. Tian et al. propose another mode of action of IGF-1 (PMID: 29131241). Please discuss.

Reviewer #2 (Remarks to the Author):

In the submitted manuscript titled " Macrophage-derived IGF-1 protects the neonatal intestine against necrotizing enterocolitis by promoting microvascular development", Yan et al build upon previous work from the De Plaen lab examining vascular development and its contribution to necrotizing enterocolitis (NEC) or NEC prevention. In this study, the authors provide a nice demonstration of the role of macrophage-derived IGF1 in endothelial cell proliferation. They demonstrate that serum and intestinal IGF1 levels are lowest in the first few days of life in mouse pups and that IGF1 levels are also low in mice subjected to an experimental NEC model. They then use neonatal, murine endothelial, and macrophage co-culture to demonstrate that intestinal macrophages promote endothelial cell proliferation and sprouting via IGF1. They use the IGF1 receptor inhibitor PPP to block IGF1 signaling and show that the endothelial cell sprouting effects from macrophage co-culture are lost. The authors then use CX3CR1-Cre to deplete IGF1 in

macrophages in vivo and demonstrate that this leads to shorter intestines, modest reductions in endothelial cell proliferation and overall endothelial cell number, and reduced expression of VEGF164 protein. In vitro macrophages with reduced IGF1 levels fail to induce normal endothelial cell sprouting. Mice harboring macrophages with diminished IGF1 are more susceptible to experimental NEC. The authors also show that blocking IGF1 in vivo results in reduced endothelial cell proliferation similar to that seen in NEC and that exogenous treatment with IGF1 enhances VEGF expression, rescues endothelial cell proliferation, improves vessel density, and enhances survival in experimental NEC. Lastly, the authors show that IGF1-expressing macrophages and endothelial cell proliferation are both reduced in human NEC compared to non-NEC controls. This work is novel and will be of interest to those who study and treat necrotizing enterocolitis, but also relevant to the fields of intestinal and immune development.

Strengths

1. The question is novel and addressed by well-designed and clearly presented experiments.
2. This study is well powered with sufficient replicates and repeat experiments.
3. Experiments were conducted in vitro, in vivo, and findings were validated in human patient samples, which adds to the translational relevance of the work.
4. The authors include all source data and clear explanations of data analyses and methods.
5. The methods are sufficiently detailed for others to reproduce the work.

Suggested improvements

1. The manuscript would benefit from a summary figure or graphical abstract.
2. Treatment with exogenous IGF1 will affect the intestinal epithelial cells and could also contribute to enhanced survival in the NEC model. This should be mentioned in the discussion.

Reviewed by Sarah Andres, PhD

Reviewer #3 (Remarks to the Author):

This is a well written and well performed study that sheds light on some of the novel and previously unexplored mechanisms by which vascular development occurs under the regulation of macrophage IGF1, and how disruption in the vascular developmental processes can lead to NEC. The studies are well described and the data are convincing. The inclusion of mouse and human data is a strength. The combination of in vitro and in vivo data adds to the validity of the proposed mechanisms. Overall, the work is important and the authors are to be congratulated for a superb study.

I have the following questions:

1. it is interesting that there are two nadirs of igf1 and igf-1r in intestinal tissue in mice. is the second nadir related to weaning from breast milk? Could this be tested by supplementing breast milk to mice and assessing whether both levels persist? If so, this finding would direct attention to additional factors that regulate the expression of these important proteins.
2. the co-association of macrophages and endothelial cells is noteworthy. Could this be quantified using image analysis software please. Is the colocalization 100% or less and is it affected by NEC?
3. cx3cr1 will drive development of lamina propria and muscularis macrophages in the neonatal gut. Could the authors please confirm that they have selective depletion of lamina propria macrophages by immunostaining or flow cytometry.

Subject: Manuscript COMMSBIO-21-2122-T

Point-by-Point Response to Reviewers' Comments

Reviewer #1:

Critique 1a. *“In general, I would recommend improving the phenotype description of the cells of interest. E.g., I wonder whether using CX3CR1 as single marker for the isolation/identification of lamina propria macrophages is sufficient, as CX3CR1 also is expressed in monocytes, NK cells and dendritic cells PMID: 10805752; PMID: 15653504). At least for flow cytometry analyses it would be better to use a combination of more specific markers (e.g., F4/80+, CD11b+, Ly6G- etc.) or show functionally that the majority of cells purified from murine intestinal tissue as described in the methods section indeed are macrophages and for instance not dendritic cells.”*

RE: To confirm that CX3CR1 cells in the neonatal mouse intestine are indeed macrophages, we ran flow cytometry studies using CX3CR1-GFP expressing pups and stained for CD45, CD11b, NK1.1, F4/80, Ly6C, CD64, CD11c, Siglec F, Ly6G, MHCII and GFP. The results are now shown in supplemental Figure 3:

When neonatal mouse intestinal lamina propria live, single, CD45⁺ cells were gated and GFP⁺ (CX3CR1⁺) (light blue) and GFP⁻ (CX3CR1⁻) (red) populations were analyzed for the NK cell marker NK1.1, the vast majority of the neonatal intestinal NK cells were CX3CR1⁻ (Suppl. Fig. 3a2). Also, most CX3CR1 positive cells were F4/80⁺ (Suppl. Fig. 3a3). (Cx3cr1⁻ CD11b⁺ such as eosinophils and dendritic cells can also be F4/80⁺ (Suppl. Fig. 3a4)¹. After selecting for live, single, CD45⁺ CD11b⁺ SiglecF⁻ Ly6G⁻ cells, neonatal intestinal CX3CR1⁺ (light blue) and CX3CR1⁻ cells (red) were analyzed, the vast majority of the neonatal intestinal LyC6⁺ cells (monocytes) were CX3CR1⁻ (Suppl. Fig. 3b2). Also, most CX3CR1 positive cells were CD64⁺ (macrophages) (Suppl. Fig. 3b3). Alternatively, neonatal intestinal live, single, CD45⁺ CD11b⁺ SiglecF⁻ Ly6G⁻ cells were gated to show the macrophages (CD64⁺), monocytes (CD64⁻Ly6C⁺ MHCII⁻) and dendritic cells (CD64⁻Ly6C⁻ MHCII⁺). While as expected the macrophages (CD64⁺ cells) strongly expressed CX3CR1⁺ (light blue) (Suppl. Fig. 3c1-3), the dendritic cells (CD64⁻Ly6C⁻ MHCII⁺) (light green) did not (Suppl. Fig. 3c1-4). We have now added on page 9 the following sentence: “Indeed, in the neonatal intestine, CX3CR1⁺ cells (macrophages) express CD11b, F4/80, CD64, but were negative for Ly6C, CD11c and NK1.1 (Suppl. Fig 3).”

Critique 1b. *“In addition to this, I recommend to provide a full gating strategy for all FACS analyses (in the supplement) including a detailed description of how the different cell types were identified. Data presented for instance in Fig. 3e otherwise might be misleading as it looks like there are no Tie-2 negative cells at all present in the co-culture.”*

RE: Thank you. The full gating strategy for our flow cytometry studies is now included in supplemental figure 2.

Critique 2. *“A description is missing of how epithelial cells, macrophages and endothelial cells were isolated/purified from murine tissue to generate protein lysates for the western blot analysis shown in Fig. 2d. In the methods section on western blotting, only the use of small intestine lysates is mentioned. For better understanding and conclusiveness, I suggest preparing an extra paragraph in the methods section to describe cell isolation procedures from murine intestinal tissue for respective assays used in this work.”*

This is now included in the method section in a new section intitled “Cell Isolation” on page 20.

Critique 3a. “The benefit of Fig. 3d is unclear and not explained in the text (no novel information for the readership). I would recommend omitting it.”

RE: As suggested by this reviewer, we have now omitted the old Fig 3d.

Critique 3b. “In addition, the results shown in Fig. 5h do not seem to be very meaningful considering the results regarding intestine length shown in Suppl. Fig. 1c. You would expect a lower cell number if the intestine is significantly shorter in pups with IGF-1-deficient macrophages. I would recommend using relative values (i.e. relative to the isolated overall cell number) for comparison of the two groups.”

Thank you for your suggestion. We have now replaced the old Fig 5h with a new figure showing the endothelial cell number relative to the isolated total cell number per intestine normalized to the mean value of *Igf1^{fl/fl}* group.

Critique 4. With their work, authors contribute to elucidate a potential mechanism by which IGF-1 protects from NEC. Tian et al. propose another mode of action of IGF-1 (PMID: 29131241). Please discuss.

RE: we have now included in the discussion the following paragraph: “Beside its protective effect on the microvasculature, IGF-1 has also been shown to protect intestinal epithelial cells from oxidative stress-induced apoptosis². Also, enterally administered IGF-1 has been shown to down-regulate TLR4 signaling, reducing the inflammatory response and to protect against intestinal injury in a rat NEC model³. Together, this suggests that the protective effect of IGF-1 in the neonatal intestine occurs at several levels.”

Reviewer #2:

Critique 1. The manuscript would benefit from a summary figure or graphical abstract. - Our format supports summary figures but not graphical abstracts.

RE: We have now added the summary figure as figure 9.

Critique 2. Treatment with exogenous IGF1 will affect the intestinal epithelial cells and could also contribute to enhanced survival in the NEC model. This should be mentioned in the discussion.

RE: Thank you for your thoughtful comment. We have now included in the discussion the following paragraph: “Beside its protective effect on the microvasculature, IGF-1 has also been

shown to protect intestinal epithelial cells from oxidative stress-induced apoptosis². Also, enterally administered IGF-1 has been shown to down-regulate TLR4 signaling, reducing the inflammatory response and to protect against intestinal injury in a rat NEC model³. Together, this suggests that the protective effect of IGF-1 in the neonatal intestine occurs at several levels.”

Reviewer #3:

Critique 1. it is interesting that there are two nadirs of igf1 and igf-1r in intestinal tissue in mice. is the second nadir related to weaning from breast milk? Could this be tested by supplementing breast milk to mice and assessing whether both levels persist? If so, this finding would direct attention to additional factors that regulate the expression of these important proteins.

RE: Thank you for your thoughtful comment. While we don't think this can be easily tested, we agree with this reviewer that the second nadirs of IGF-1 and IGF-1R could be due to both weaning from breast milk and lower concentrations of IGF-1 in the breastmilk. While it is very difficult to be monitored in mice, the concentration of IGF-1 in human milk has been shown to decline over the first 6 months of life⁴. We have now added the following sentence in the discussion: “Interestingly both IGF-1 and IGF-1R, after a sharp decrease immediately after birth, are highly expressed in the mouse intestine by the end of first week of life. Both IGF-1 and IGF-1R significantly decrease between 7 and 21 days of age, which corresponds to the weaning period. As breast milk contains significant amount of IGF-1⁵, it may play an important role in promoting IGF-1 signaling in the intestine during this critical period and further studies are needed in this regard.”

Critique 2. the co-association of macrophages and endothelial cells is noteworthy. Could this be quantified using image analysis software please. Is the colocalization 100% or less and is it affected by NEC?

RE: Indeed, we noted significant intertwining of macrophage protrusions with endothelial cells in the intestinal villi of dam fed pups which was less apparent in the villi of NEC pups. However, the co-association of endothelial cells and macrophages was difficult to quantify. Instead, we were able to quantify the observed macrophage morphology changes which include a decrease in macrophage relative cellular area and perimeter, and an increase in macrophage circularity in the villi of pups exposed to the NEC model compared to the dam fed controls (please see new supplemental Figure 1). These could in part be F4/80⁺ monocytes that are recruited to the intestine tissue during NEC/inflammation.

Critique 3. cx3cr1 will drive development of lamina propria and muscularis macrophages in the neonatal gut. Could the authors please confirm that they have selective depletion of lamina propria macrophages by immunostaining or flow cytometry.

RE: While we confirmed depletion of IGF-1 in isolated CX3CR1⁺ cells obtained from intestinal lamina propria preparation in macrophage-deficient IGF-1 mice using western blot (Fig 5b) and we showed, using CX3CR1-GFP mice, that CX3CR1 is specifically expressed by macrophages, but not other inflammatory cells examined (cfr. New supplemental figure 3 - Response to reviewer 1-1a), we were unable to immunolocalize CX3CR1 in this experiment. Indeed, given

the lack of suitable efficacious anti-CX3CR1 antibody for IF in mice, we were unable to identify CX3CR1⁺ cells by immunofluorescence in our mouse tissues to address their specific localization lamina propria vs. muscularis. We have added this limitation of our study in the discussion on page 15 as follows: “*However, our study could not differentiate an effect arising from the macrophages derived from the lamina propria vs. the muscularis mucosa.*”

REFERENCES:

- [1] Lokka E, Lintukorpi L, Cisneros-Montalvo S, Mäkelä JA, Tyystjärvi S, Ojasalo V, Gerke H, Toppari J, Rantakari P, Salmi M: Generation, localization and functions of macrophages during the development of testis. *Nat Commun* 2020, 11:4375.
- [2] Baregamian N, Song J, Jeschke MG, Evers BM, Chung DH: IGF-1 protects intestinal epithelial cells from oxidative stress-induced apoptosis. *J Surg Res* 2006, 136:31-7.
- [3] Tian F, Liu GR, Li N, Yuan G: Insulin-like growth factor I reduces the occurrence of necrotizing enterocolitis by reducing inflammatory response and protecting intestinal mucosal barrier in neonatal rats model. *Eur Rev Med Pharmacol Sci* 2017, 21:4711-9.
- [4] Milsom SR, Blum WF, Gunn AJ: Temporal changes in insulin-like growth factors I and II and in insulin-like growth factor binding proteins 1, 2, and 3 in human milk. *Horm Res* 2008, 69:307-11.
- [5] York DJ, Smazal AL, Robinson DT, De Plaen IG: Human Milk Growth Factors and Their Role in NEC Prevention: A Narrative Review. *Nutrients* 2021, 13:3751.

Reviewer #1 (Remarks to the Author):

I thank the authors for their response to our comments. I appreciate that all points have been addressed and revised in the manuscript. However, I have some concerns regarding the new Suppl. Fig. 2 and 3:

Ad Suppl. Fig. 2:

The gating strategy is not described in the figure legend.

Ad Suppl. Fig. 3:

By additional FACS analyses, authors could confirm that the majority, but not all Cx3cr1+ cells are macrophages. There are Cx3cr1+ cells that e.g. do not express CD11b (Suppl. Fig. 3a2) and some monocytes seem to express intermediate levels of Cx3cr1 (Suppl. Fig 3c3). To claim that the major proportion of Cx3cr1+ cells are indeed macrophages, gates and percentages would need to be shown. These are missing, alongside with the complete gating strategy for these experiments. I would recommend omitting the Cx3cr1- cells in Suppl. Fig. 3a2-4 + 3b2-4. Instead, focus on quantifying the macrophage-population within the Cx3cr1+ cells.

Reviewer #2 (Remarks to the Author):

The authors have sufficiently addressed my comments and incorporated the feedback of all three reviewers.

Reviewed by Sarah Andres, PhD

Reviewer #3 (Remarks to the Author):

my comments were addressed
congrats on a nice study

Subject: Manuscript COMMSBIO-21-2122-T

Point-by-Point Response to Reviewers' Comments

Reviewer #1:

Ad Suppl. Fig. 2:

The gating strategy is not described in the figure legend.

RE: Thank you for pointing that out. We have now described the gating strategy in the figure legend.

Ad Suppl. Fig. 3:

By additional FACS analyses, authors could confirm that the majority, but not all Cx3cr1+ cells are macrophages. There are Cx3cr1+ cells that e.g. do not express CD11b (Suppl. Fig. 3a2) and some monocytes seem to express intermediate levels of Cx3cr1 (Suppl. Fig 3c3). To claim that the major proportion of Cx3cr1+ cells are indeed macrophages, gates and percentages would need to be shown. These are missing, alongside with the complete gating strategy for these experiments. I would recommend omitting the Cx3cr1- cells in Suppl. Fig. 3a2-4 + 3b2-4. Instead, focus on quantifying the macrophage-population within the Cx3cr1+ cells.

RE: Thank you for your suggestions. We know based on our FMO staining that our GFP gate is at the correct position and the relatively high negative levels are due to the high autofluorescence of neonatal lamina propria cells. We have now replaced Supplemental Fig. 3 with a clearer FACS analysis image (with complete gating strategy) that shows more clearly that the majority of Cx3cr1⁺ cells are indeed macrophages.